# Genome-Wide Identification of the *DOF* Gene Family Involved in Fruitlet Abscission in *Areca catechu* L.

**DOI:** 10.3390/ijms231911768

**Published:** 2022-10-04

**Authors:** Jia Li, Xiaocheng Jia, Yaodong Yang, Yunche Chen, Linkai Wang, Liyun Liu, Meng Li

**Affiliations:** 1Coconut Research Institute, Chinese Academy of Tropical Agricultural Sciences, Wenchang 571339, China; 2College of Life Sciences, Chongqing Normal University, Chongqing 401331, China; 3College of Life Science and Technology, Central South University of Forestry and Technology, Changsha 410004, China

**Keywords:** *Areca catechu* L., fruitlet abscission, *DOF* gene, expression profile, lignin

## Abstract

Fruitlet abscission frequently occurs in *Areca catechu* L. and causes considerable production loss. However, the inducement mechanism of fruitlet abscission remains mysterious. In this study, we observed that the cell architecture in the abscission zone (AZ) was distinct with surrounding tissues, and varied obviously before and after abscission. Transcriptome analysis of the “about-to-abscise” and “non-abscised” AZs were performed in *A. catechu*, and the genes encoding the plant-specific DOF (DNA-binding with one finger) transcription factors showed a uniform up-regulation in AZ, suggesting a role of the DOF transcription in *A. catechu* fruitlet abscission. In total, 36 members of the *DOF* gene family distributed in 13 chromosomes were identified from the *A. catechu* genome. The 36 *AcDOF* genes were classified into nine subgroups based on phylogenic analysis. Six of them showed an AZ-specific expression pattern, and their expression levels varied according to the abscission process. In total, nine types of phytohormone response *cis*-elements and five types of abiotic stress related *cis*-elements were identified in the promoter regions of the *AcDOF* genes. In addition, histochemical staining showed that lignin accumulation of vascular bundles in AZ was significantly lower than that in pedicel and mesocarp, indicating the specific characteristics of the cell architecture in AZ. Our data suggests that the DOF transcription factors might play a role in fruitlet abscission regulation in *A. catechu*.

## 1. Introduction

*Areca catechu* L. is one of the most important tropical industrial crops. However, a large portion of *A. catechu* fruit, areca nut, will shed during the fruitlet development period, and this causes considerable production loss. According to the survey data of in the Hainan province of China, the average setting rate of areca nut is less than 12% in most *A. catechu* planting areas. Nevertheless, high-yield individuals with fruit setting rates over 50% were observed despite the low average setting rate, indicating the possibility of *A. catechu* plants to prevent fruitlet abscission.

Fruit abscission is a common phenomenon in fruit tree species. The factors determining fruit abscission are classified into two types, i.e., ontological factors and environmental cues. Fruit abscission could be induced by genetic load, nutrition deficiency, insufficient pollination and fertilization, and abnormal development of flower organs [1]. Therefore, fruit abscission could occur at both the developing and maturity stages due to different conditions. Most studies focusing on the molecular mechanism of organ abscission were performed in *Arabidopsis* and tomato (*Solanum lycopersicum*), and a lot of key genes involved in the regulation of fruit abscission were identified. For example, the gene encoding ethylene receptor and its downstream members such as ETR1 (ethylene response 1), EIN2 (ethylene-insensitive 2), EIN3 (ethylene-insensitive 3) and ERS2 (ethylene response sensor 2), etc; as well as ARF (auxin response factor) 1, ARF2, ARF7 and ARF19 were verified to participate in the regulation of floral organ abscission in *Arabidopsis* [2,3]. Thus far, at least nine gene families encoding transcription factors have been proposed to regulate organ abscission, including the *Aux/IAA* (Auxin/Indole-3-Acetic Acid), *ARF*, *EIN3* and *ERF* (ethylene- responsive factor) involved in the plant hormone signal pathway [2,3,4,5,6], the *MADS-box* genes involved in organ development and abscission zone (AZ) formation [7,8,9], and a number of members in the *KNOTTED-like homeobox*, *HD-ZIP* (Homeodomain-leucine zipper), *DOF* (DNA binding with one finger) and *ZFP* (zinc finger transcription factor) gene families [5,10,11,12,13]. This evidence indicated that fruit abscission is under the complicated control of the crosstalk of plant hormone signals, molecular regulators and environmental cues.

To reveal the molecular mechanism of fruitlet abscission in *A. catechu*, we performed RNA-seq analysis in areca nut samples at different stages during fruitlet abscission, and noticed that almost all members of the *DOF* gene family showed a significantly up-regulation in AZ and shedding fruits, indicating that this gene family plays a critical role in fruitlet abscission of *A. catechu*. The encoding products of the *DOF* genes are plant-specific transcription factors. They belong to the zinc finger super family, which is generally composed of 200–400 amino acids [14,15]. The DOF domain is named after a conserved DNA binding domain rich in cysteine residues. The N-terminal binding domain of the DOF protein is a conserved C2-C2 type single zinc finger domain composed of 52 amino acid residues in which there are four conserved Cys residues and one covalently bound Zn^2+^ [16,17]. The DOF protein regulates the expression of downstream genes by binding its N-terminal DOF domain to their promoter regions [15]. Almost all DOF proteins bind to the promoter region via the core *cis*-element AAAG in a sequence-specific manner, thus any mutation occurring in the core element will change the capacity of the DOF domain to recognize specific *cis*-elements [16,18,19]. In addition to the capacity of DNA binding, the DOF proteins were also reported to interact with the bZIP and GAMYB transcription factor at the protein-protein level [19,20,21].

Recent studies have shown that the DOF transcription factors could participate in plant growth, organ development and responses to various environmental stresses. The *DOF* genes in *Arabidopsis* have been proved to be involved in seed germination and the photoperiod control of flowering [22], stomatogenesis [23], vascular development [24], abiotic stress tolerance [15,22], as well as organ abscission [11,25]. Furthermore, a number of *DOF* genes with different functions were identified from rice (*Oryza sativa*). For examples, the *OsDOF24* and *OsDOF25* gene altered the expression of the *GluB-1* gene encoding a storage protein, and thus regulated the storage protein composition in rice seeds [26]. Overexpression of an *OsDOF12* gene inhibited the growth of main and lateral branches, decreased plant height, shortened upright leaves and reduced spikelet length in transgenic rice plants [27], while, the *OsDOF18* gene was reported to control ammonia uptake by inducing the activity of ammonia transporters in rice roots [28]. Overexpression of a *SlCDF3* gene from tomato, which regulates the expression of genes related to redox homeostasis, photosynthetic performance and primary metabolism, increased the biomass of transgenic tomato plants by improving photosynthesis and sugar utilization [29].

Although the *DOF* gene family has been verified to be involved in diverse biological processes in plants, its function on organ abscission is rarely reported. A case reported in *Arabidopsis* demonstrated that the *PGAZAT* (polygalacturonase) gene is under the control of the AtDOFs. The *PGAZAT* gene determines cell wall degradation, and the polygalacturonase activity could be inhibited by the AtDOF4 transcription factor or be enhanced by another transcription factor, AtDOF4.7. Thus, overexpression of the *AtDOF4* or *AtDOF4.7* gene eventually resulted in no shedding and early abscission in transgenic *Arabidopsis*, respectively [11,12,25]. This evidence indicates the potential role the *DOF* gene plays in organ abscission, as well as the fine division of the function of the DOF family. In order to identify the *DOF* genes involved in fruitlet abscission of *A. catechu*, we performed a genome-wide screening of the *AcDOF* gene family. The chromosome distribution, conserved motifs, phylogenetic relationships and *cis*-elements of the promoter region of these *AcDOF* genes were analyzed to elucidate their functions. The results will provide new insights for the DOF pathway functioning in the fruitlet abscission process of *A. catechu*.

## 2. Results

### 2.1. Dynamic Observation of the Process of Fruitlet Abscission in A. catechu

*A. catechu* is a monoecious and cross pollinated plant. The spadix of *A. catechu* is visible after the dehiscence of surrounding bracts (Figure 1A). The male flowers will gradually open 3–11 days after bract dehiscence and then start to shed. The female flowers will open when 70–100% of the male flower abscised. A small part of female flowers will drop off before bloom (Figure 1B), while most female flowers will shed after pollination (Figure 1C). The fruitlet abscission data collected in this study are on the ovaries of opened or pollinated female flowers. The abscission rate was investigated from 2020 to 2021. Based on the field data, there is a peak period of fruitlet abscission during the four-month developmental process of *A. catechu* fruits (Figure 1A–F). The fruitlet shedding starts on the 10th day after the female flowers bloom. Subsequently, the abscission rate reaches 21.8% and 74.8% on the 21st and 28th day, respectively (Figure 1G). However, the abscission rate in the preserved fruits after this stage did not increase significantly, indicating that the fruitlet abscission signals were enriched in the first month after pollination.

To observe the cell architecture of AZ in *A. catechu*, the junction area between fruit and pedicel was dissected longitudinally by paraffin section. The cells in AZ were small, dense and regularly arranged, which makes them easy to distinguish from other cells (Figure 2A). An obvious detached region was observed in AZ during the “about-to-abscise” period (Figure 2B). The cells in the non-abscised part of AZ were round or oval, smaller than the adjacent cells, and with dense cytoplasms, while, the cells in the abscised part of AZ were longitudinally oval and single layer arranged with larger gaps between them. In addition, there is less sediment in the intercellular cell space in the non-abscised part of AZ, while after abscission, the AZ cells were dark stained in the intercellular space (Figure 2C–F). This variation might be due to the accumulation of secretions related to cell wall biosynthesis in this area, which may form a protective layer after abscission. Furthermore, a number of differentiated and lignified vascular bundles were observed in the pedicel and mesocarp, while, the lignification degree of vascular bundles decreased in AZ, and most vascular bundles were interrupted after abscission (Figure 2A,C,E). At the same time, dividing cells were observed in the non-abscised part of AZ, while the cell division in the detached part of AZ is no longer obvious (Figure 2B,D,F).

### 2.2. Transcriptome Analysis Identified Differentially Expressed DOF Genes in A. catechu

To reveal the molecular mechanism underlying fruitlet abscission in *A. catechu*, we performed transcriptome analysis among AZ samples at different stages. The transcriptome data has been deposited into the China National Center for Bioinformation with the code CRA007290. A number of differentially expressed genes (DEGs) were identified during the abscission process. 

In total, 94 genes encoding transcription factors (TF) were identified as DEGs across different samples. Twenty one of the differential TFs are closely related to phytohormone response, and 73 of them are classified into different TF families, including bHLH (basic helix-loop-helix protein), bZIP (basic region/leucine zipper motif), GRAS (chitin-inducible gibberellin-responsive protein), ZF (zinc finger protein), KNOX (knotted1-like homebox), MADS-box (*MCM1*, *AGAMOUS*, *DEFICIENS* and *SRF*), MYB, NAC (nam, ataf1/2, and cuc2) and WRKY. We noticed that 49 genes encoding TFs were up-regulated in AZ, including most members of *MYB*, *NAC* and *ZF*; while 45 genes were down-regulated, including *ERFs* and *ARFs* (Figure 3A, Appendix A).

Among them, we noticed that genes encoding the DOF transcription factor belonging to the ZF family showed obvious changes in expression level before and after fruitlet abscission, indicating that the DOF transcription factors were involved in this process (Figure 3B). In total, 25 *AcDOF* genes were specifically expressed in AZ, and six of them showed significantly different expression (False Discovery Rate (FDR) < 0.05 and |log_2_ Fold Change| ≥ 1) between the “about-to-abscise” and “non-abscised” parts of AZ, including *AcDOF1*, *AcDOF2*, *AcDOF3*, *AcDOF4*, *AcDOF8* and *AcDOF9*.

### 2.3. Expression Profiles of the AcDOF Genes

To evaluate the functions of the *AcDOF* genes in fruit growth and development of *A. catechu*, the expression levels of 6 *AcDOF* genes in different tissues (AZs, leaves, roots, male flowers, calyx, petals and ovaries) were detected with qRT-PCR. The results showed that nearly all six *AcDOF* genes exhibited relatively high expression levels in AZ (Figure 4). We further analyzed the expression patterns of these 6 *AcDOF* genes during fruitlet abscission. The *AcDOF4*, *AcDOF8* and *AcDOF9* genes showed similar expression patterns. Their expression levels reached the highest on the 21st day after female flowers opened, and then decreased. For the *AcDOF1* and *AcDOF3* genes, their expression levels kept increasing after female flowers opened, reached a peak on the 28th day, and then started to decrease. The expression levels of all the 6 *AcDOF* genes declined to the bottom on the 35th day (Figure 5). The expression patterns of these *AcDOF* genes were consistent with the fluctuation of the abscission rate, suggesting their roles in the regulation of fruitlet abscission. In addition, the *AcDOF1*, *AcDOF2*, *AcDOF3*, *AcDOF8* and *AcDOF9* genes were highly expressed in male flowers and leaves (Figure 4), while, the *AcDOF4* gene was specifically expressed in floral organs, indicating that the *AcDOF* genes were also involved in the growth or development regulation in these organs.

### 2.4. Genome-Wide Identification of the DOF Genes in A. catechu

The transcriptome analysis results demonstrated that the *DOF* genes play an important role in fruitlet abscission in *A. catechu*. However, we questioned whether there are other *DOF* genes of *A. catechu* that also participate in the regulation of abscission. Therefore, based on the *A. catechu* genome data [30], a total of 36 *AcDOF* genes were identified. For the AcDOF proteins, the amino acid length ranges from 189 to 518, the predicted molecular weight ranges from 20.61 kDa to 56.23 kDa, and the isoelectric point ranges from 5.16 to 9.41 (Table 1). The subcellular localization of the AcDOF proteins were predicted using PSORT. The results indicated that most AcDOF proteins were predicted to locate in the nucleus except AcDOF12 and AcDOF28, which were predicted to locate in the mitochondria.

The number of exons and introns of the *AcDOF* genes ranges from one to two and zero to one, respectively. There are 13 *AcDOF* genes, each of which has only one exon, and 20 *AcDOF* genes, each of which contains two exons. An exception is *AcDOF28*, which possesses 10 exons and nine introns. The arrangement of exons and introns is relatively conservative in the *AcDOF* members from the same subgroup. In addition, large introns were found in *AcDOF1*, *AcDOF27*, *AcDOF28* and *AcDOF34* (Figure 6).

The distribution of the *AcDOF* genes on chromosomes was analyzed by searching the *AcDOF* gene sequences against the *A. catechu* genome. The results demonstrated that the 36 *AcDOF* genes were distributed on 13 of the 16 chromosomes of the *A. catechu* genome, and the distribution was uneven. There are five, five, four and four *AcDOF* genes that were located on chromosome1, chromosome15, chromosome3, and chromosome16, respectively. The other *AcDOF* genes were distributed on chromosome 2, 4, 5, 6, 8, 9, 10, 11 and 13 (Figure 7).

In order to further understand the evolutionary mechanisms of the AcDOF family, we built three comparative syntenic maps of *A.catechu* with three other special species, including two species of monocots, rice (*Oryza sativa*) and oil palm (*Elaeis guineensis*), and one species of dicot, *Arabidopsis* (Figure 8). A total of 83, 26 and 32 orthologous gene pairs were found between the *AcDOFs* and other genes in oil palm, rice and *Arabidopsis*, respectively. Some *AcDOF* genes were associated with at least four syntenic gene pairs, such as *AcDOF29*, *AcDOF30* and *AcDOF31*, and these genes might play important roles in the *DOF* gene family during evolution.

### 2.5. Conserved Domains of AcDOF Proteins and Phylogenetic Analysis

The DOF-DNA domain, which is usually located near the N-terminal region of the protein, is a conserved domain for the DOF proteins. The DOF-DNA domain was found in all the AcDOF proteins through MEME tools, which is consistent with the results in *Arabidopsis*. Each DOF-DNA domain of the AcDOF protein contains approximately 56 amino acid residues (motif 1, as shown in Figure 9). Besides motif 1, there are another nine motifs that were identified in both the AcDOF and AtDOF proteins. Some motifs were specifically present in individual groups. For example, motif 9 was only found in group I, motif 4 and 7 were only found in group VI, while all AcDOF proteins in group VI contain motif 2, 3, 5 and 10. Details on these motif features are shown in Figure 9. The Zn finger-like structure is the string CX_2_CX_21_CX_2_C type, which binds zinc (Zn^2+^).

The tree was constructed from a complete alignment of 36 *Arabidopsis* and 36 *A. catechu* DOF proteins using the maximum likelihood method; bootstrap values of greater than 50% are shown at the nodal branches. The right portion shows the distribution of conserved motifs in *Arabidopsis* and *A. catechu* DOF proteins. 

Phylogenetic analysis of AcDOF proteins and AtDOF proteins from *Arabidopsis* was performed using the ML method to clarify the evolutionary relationship between the AcDOF proteins and the DOF proteins from other species. The DOF proteins from different species was divided into four main lineal homologous groups (A, B, C and D) and nine branches, which are similar to the results reported by Zhou et al. [31]. Among them, the group D constitutes the largest branches, including 11 members (30.55%). There are nine, nine, and seven members in group C, group B, and group A, respectively. The C2.1 branch contains four and five DOF members from *A. catechu* and *Arabidopsis*, respectively, but they are obviously separated. In addition, no AcDOF protein was found in the C3 branch, suggesting the lack of the C3 branch in the *A. catechu* genome.

### 2.6. Cis-Elements Analysis on the Promoter Regions of the AcDOF Genes

In order to elucidate the regulating patterns of the *AcDOF* genes, *cis*-elements in the promoter regions (2000 bp sequences upstream of the translation start site) of the 36 *AcDOF* genes were analyzed using PlantCARE. Five stress-related *cis*-elements were identified as the most abundant in the promoter regions of the *AcDOF* genes. Among the *cis*-elements identified related to stress response, the antioxidant response elements are the most abundant, and the promoter regions of 27 *AcDOF* genes contain this element. Other four stress-related *cis*-elements, including W-box, WUN-motif, MBS and TC-rich repeats, were found in 18, seven, nine, and eight promoter regions of the *AcDOF* genes, respectively. Furthermore, various hormone-related *cis*-elements were also identified in the promoter regions of the *AcDOF* genes, including abscisic acid (ABA)-responsive element (ABRE), ethylene-responsive element (ERE), salicylic acid (SA)-responsive element (TCA-element), methyl jasmonate (MeJA)-responsive element (CGTCA-motif), auxin-responsive elements (AuxRR-core and TGA-element), and gibberellin-responsive elements (P-box, GARE-motif and TATC-box) (Figure 10). These findings indicated that the *AcDOF* genes might be regulated by various stress responses and hormone signaling.

### 2.7. Prediction of Downstream Genes Regulated by the AcDOF Transcription Factor

Since the DOF protein is a putative transcription factor, we performed the prediction on *cis*-elements that can be recognized and bound by AcDOF4, which was selected as a representative, through a TF (transcription factor)-centered Yeast One Hybrid technique. In total, 33 motifs were identified as candidate targets of AcDOF4 (Appendix A). The presence of this motif in the promoter regions of some well-documented genes involved in abscission, including genes encoding CesA (cellulose synthase A catalytic subunit 5), PE/PEI (pectinesterase/pectinesterase inhibitor 41), EXP2 (expansin 2), LRX7 (leucine-rich repeat extensin-like protein 7), Csl (cellulose synthase-like protein), PG (Polygalacturonase) and GLU (glucan endo-1,3-beta-glucosidase) was screened. The results demonstrated that all analyzed genes contain at least one candidate motif recognized by AcDOF4 in their promoter regions, while most motifs are unnamed with unknown functions (Table 2). Consistent with the prediction, most of these genes showed significantly different expressions between the “about-to-abscise” and “non-abscised” parts of AZ (Figure 11A), and the expression data of some candidate genes was confirmed by qPCR as well (Figure 11B).

## 3. Discussion

The specific expression pattern of some key genes in plant tissues and developmental stages can provide important information about the gene functions [32,33]. In this study, six *AcDOF* genes that showed significant differential expression during the fruitlet abscission process in *A. catechu* were identified based on the transcriptome data. The expression patterns of these six genes showed an AZ-specific manner and have a positive correlation with the abscission rate. The tissue-specific expression patterns of the *DOF* genes were also reported in other plant species. For example, the *ZmDOF3* gene was only expressed in maize endosperm to regulate starch accumulation and aleurone layer development [34]. Another maize *DOF* gene, *ZmDOF36*, was highly expressed in maize endosperm, and participated in starch synthesis by regulating the expression of starch synthesis genes [35]. Remarkably, the *AtDOF4.7* gene in *Arabidopsis* is specifically expressed in flowers and young siliques to directly inhibit the expression of *PGAZAT*, a polygalacturonase gene related to cell wall degradation, thereby affecting cell wall degradation, resulting in the failure of flower organs to fall off normally [11,25]. This finding indicates the role of the DOF genes playing in the fruit or flower abscission process. Similarly, the *AcDOF4* gene identified in this study was only expressed in flower organs and fruit abscission layers. The expression level of the *AcDOF4* gene reached the highest on the 21st day, when the fruitlet abscission reached a peak (Figure 5). The *AtDOF4.7* gene was specifically expressed in flower organs and pods, and maintained a high expression level during the process of petal abscission. However, its expression level decreased significantly when abscission had completed.

A large number of differentiated and lignified vascular bundles were observed in the pedicel and mesocarp (Figure 2A,C,E), but the degree of lignification of vascular bundles in the AZ was obviously decreased (Figure 2B,D,F). Less lignification and differentiation in vascular cells of AZ makes them more vulnerable to pectinase hydrolysis, therefore easier to shed [36]. This observation indicated that the cell wall biosynthesis and differentiation of AZ cells were specifically controlled. In oil palm, lignin accumulation in the vascular cells of AZ was much less than that in the adjacent mesocarp and pedicel tissues, indicating that lignin synthesis was inhibited in the AZ cell layer [37]. The fact that spatial lignification difference of the AZ cell determined the functional specificity of the AZ cell layer was also reported in *Arabidopsis* [38,39]. Studies of abscission on dicotyledon model plants showed that the development and function of AZ are strictly regulated [40]. For example, the differentiation, location and function of AZ in *Arabidopsis* were under the control of transcription factors blade-on-petiole1/2 (BOP1/2), asymmetric leaves1 (AS1) and Hawaiian skirt (HES) [41,42,43]. While in tomato, the differentiation of the AZ in flower and fruit was controlled by completely different transcription complexes, including MADS box transcription factors and SlERF52 [7,8,44]. Although the reported regulators involved in AZ formation are different in species, their function modes are similar. The interruption of AZ development will promote cell separation and eventually lead to organ abscission. At the beginning of fruitlet abscission in *A. catechu* (Figure 3B), there was less staining in the intercellular space of the abscission layer, while after abscission, AZ cells had dark staining intercellular space (Figure 2D,F). This may be because secretions related to cell wall biosynthesis gathered in this area, which may be related to the formation of the protective layer after abscission [36].

Based on the recently released *A. catechu* genome database [30], a total of 36 *AcDOF* genes were identified (Table 1). The numbers of the *DOF* gene family in diverse plant species are close to each other. For example, there are 36 *DOF* genes in watermelon [31], 33 in pepper [18], 34 in tomato [45], 35 in potato [32], 35 in foxtail millet [46], 36 in cucumber [47], 37 in chickpea [48], and 38 in pigeonpea [49], suggesting the conservatism of the *DOF* genes and their function in plants. In order to clarify the evolutionary relationship between the *AcDOF* genes and the *DOF* genes from other species, a phylogenetic analysis was constructed based on the alignment of the DOF proteins between *Arabidopsis* and *A. catechu*. There is at least one homologous gene for each *AcDOF* gene that was found in the *Arabidopsis* genome (Figure 9). However, the AcDOF proteins were classified into nine subgroups (A, B1, B2, C1, C2.1, C2.2, C3, D1 and D2), which is consistent with those in pear [50], watermelon [31], *Arabidopsis* [51], and eggplant [52].

The expression regulation of the *AcDOF* genes was predicted by *cis*-element analysis. *Cis*-element is a noncoding DNA sequence that exists in the promoter region of a gene. The distribution of different *cis*-elements in the promoter region indicates the differences in gene regulation and function [53]. A large number of *cis*-elements related to response to the environmental stress were identified in the promoter region of the *AcDOF* genes, such as MBS and W-box. Among all *AcDOF* genes, the promoter region of 27 contains at least one or more copies of these *cis*-elements, indicating that the *AcDOF* gene expression can be induced by biotic or abiotic stress (Figure 9). In addition, a number of *cis*-elements related to response to plant hormones, such as ABA, SA, GA, MeJA, ethylene and auxin, were found in the promoter region of the *AcDOF* genes. Among them, the ABRE *cis*-element response to abscisic acid exists in almost all *AcDOF* genes. The study of the *DOF* genes in watermelon demonstrated that the *DOF* genes were strongly induced by ABA [31]; In Arabidopsis, the expression of the *AtCDF3* gene was induced by cold, drought, salt and ABA treatment. Overexpression of the *AtCDF3* gene promoted tolerance to drought, cold and osmotic stress [22]. ABA is believed to be involved in the adaptation of plants to various abiotic stresses by regulating the expression of a large number of stress-related genes [31] These results suggest that the *AcDOF* gene might also be under the control of the ABA pathway. Another identified *cis*-element in the promoter region of the *AcDOF* gene related to plant hormone is the CGTCA box, which is response to MeJA. MeJA has been reported to accelerate leaf senescence [54,55], induce bud opening [56,57], as well as to regulate abscission. The role of MeJA in promoting organ abscission was proposed by the study of JA biosynthetic mutants in *Arabidopsis*. The MeJA-related mutants, *dad1* (defective in anther dehiscence 1), *dde1* (delayed dehiscence 1) and *coi1* (coronatine inductive 1), showed obviously delayed abscission [58,59]. The mechanism of MeJA mediated abscission is unclear. However, ethylene and MeJA are thought to act on the abscission of floral organs in different ways. These results suggest that the *AcDOF* gene family may be synchronically regulated by different hormones.

## 4. Materials and Methods

### 4.1. Plant Materials

An *A. catechu* cultivar “Reyan No. 1” was used for recording the abscission rate and sampling. The 8-year-old *A. catechu* plants of “Reyan No. 1” were grown in the experimental base of the Coconut Research Institute of Chinese Academy of Tropical Agricultural Sciences. A total of 30 plants with consistent physiological state were randomly selected from the planting base to investigate the fruit abscission rate from 2020 to 2021. The fruit abscission rate (%) = cumulative fruit abscission quantity on the survey day/initial fruit abscission quantity × 100%.

The samples of AZ were collected on the 25th day after pistillate flowers bloomed. The morphological changes of AZ in *A. catechu* fruitlets were recorded. The “about-to-abscise” and “non-abscised” parts are defined as the AZ parts that will and will not shed by gentle shaking or touching. The samples of the “about-to-abscise” part of AZ was collected on the tissues 2 mm from the peel side and 2 mm from the calyx side, while the “non-abscised” part of AZ was collected on the tissues 2 mm above and below the calyx side. The collected samples, including AZs, leaves, roots, male flowers, calyx, petals and ovaries were divided into two parts for cytological observation and RNA extraction, respectively.

### 4.2. Identification of the AcDOF Genes in A. catechu Genome

The amino acid sequences of *Arabidopsis* DOF proteins (AtDOFs) were downloaded from The *Arabidopsis* Information Resource (TAIR) (http://www.arabidopsis.org accessed on 8 May 2022) database. The *A. catechu* genome sequences were downloaded from the China National GeneBank Sequence Archive (https://db.cngb.org/cnsa/ accessed on 9 May 2022) of the China National GeneBank DataBase with an accession number CNP0000517 (DOI: http://dx.doi.org/10.26036/CNP0000517 accessed on 18 May 2022). A local database of the protein sequences of *A. catechu* was established using BioEdit v7.2.6. The *DOF* genes extracted from *Arabidopsis* genome were used to identify putative *DOF* genes in *A. catechu* in the local database using BLASTp. The isoelectric point (pI) and molecular weight (Mw) of the encoding products of these genes were predicted using the ExPASy proteomic website (https://web.expasy.org/compute_pi/ accessed on 18 May 2022). The PSORT (Protein Subcellular Localization Prediction Tool) online tool was used to predict the subcellular localization of all *AcDOF* genes (https://www.genscript.com/psort.html accessed on 18 May 2022).

### 4.3. Phylogenic Analysis, Motif Composition and Chromosomal Distribution

A maximum likelihood phylogenic tree of the *DOF* gene families from *A. catechu* and *A. thaliana* was constructed using MEGA 6.06 software based on amino acid sequences of the conserved DOF domain and with 1000 bootstrap replicates for reliability [60]. The MEME tool (http://meme-suite.org/tools/meme accessed on 18 May 2022) was used to predict and analyze the conserved motifs of the AcDOF proteins with the maximum number of motifs being set as 10. The structure schematic diagrams and chromosomal distribution of the *AcDOF* genes were illustrated using the TBtools software [61]. For promoter analysis, the putative promoter sequence, which was defined as the 2000 bp region upstream to the transcription start site, for each *AcDOF* gene were extracted, and the *cis*-elements distribution in the promoter regions were analyzed using the PlantCARE tool (http://bioinformatics.psb.ugent.be/webtools/plantcare/html/ accessed on 19 May 2022).

### 4.4. Light Microscopy and Image Acquisition

Histochemical staining was performed to observe the cell architecture of the tissues during fruitlet abscission. Approximately 2 mm abscission tissue were cut longitudinally and fixed in FAA (Formalin-Aceto-Alcohol) to make paraffin sections stained with safranine O-Fast green according to the method of Xu et al. [62]. In brief, the paraffin sections were first dewaxed to water, stained with Safranin O, discolored in 50, 70, and 80% alcohol sequentially, and then stained with fast green. The sections were cleared in xylene and covered with coverslip before supplied to Nikon Eclipse E100 for microscope inspection and imaging systems (Nikon DS-U3) for image acquisition and analysis.

### 4.5. Expression Analysis of the AcDOF Genes 

In order to analyze the expression profiles of the *AcDOF* genes in AZ of *A. catechu* fruitlets, the expression levels of the *AcDOF* genes were extracted from the previously obtained transcriptome data. For expression profiling, Reads Per Kilobases per Million mapped reads (RPKM) values from RNA-seq data were log2 transformed. Expression patterns with hierarchical clustering are displayed in heat map illustrator using the TBtools software [61].

The samples for RNA extraction and qPCR detecting were collected as previously described. The samples of each stage were collected from three individuals as biological replicates. Total RNA was extracted using the E.Z.N.A. Plant RNA Kit (Omega, R6827-01, New York, NY, USA). The quality of RNA, including degradation and contamination, was monitored on 1% agarose gels. RNA concentration and integrity of the total RNA were measured using a Nano Photometer Spectrophotometer (IMPLEN, Westlake Village, CA, USA) and an Agilent 2100 Bioanalyzer (Agilent Technologies, Santa Clara, CA, USA), respectively. 

The extracted RNA was reversely transcripted into cDNA using the PrimeScript^™^ RT reagent Kit (RR047Q, TaKaRa, Osaka, Japan). The cDNA was then 10 × diluted and used as templates for qPCR. The qPCR reaction system adopted the PowerUp™ SYBR™ Green Master Mix (A25777, Applied Biosystems, Waltham, MA, USA) and the qPCR program was performed on an ABI real-time instrument (QuantStudio^TM^ 6 Flex System, Waltham, MA, USA). An *A. catechu* gene, *AcActin* (CL9155.Contig7), was used as the reference gene for data normalization. Primers used in qPCR are shown in Appendix A. The relative expression level of each sample was calculated by its C_T_ value normalized to the C_T_ value of reference gene using the 2^−ΔΔCT^ method [63], and the result of each sample was obtained from three independent biological replicates with technical replicates. Data were statistically analyzed by one-way ANOVA using SPSS 20.0 software, and Tukey’s multiple range tests were used to detect significant treatment differences (*p* < 0.05).

### 4.6. TF (Transcription Factor)-Centered Yeast One Hybrid Assays

A 7 bp random motif library from yeast strain Y187 was purchased from Nanjing Ruiyuan Biotechnology Co., Ltd. (Nanjing, China). A TF-centered Y1H assay was performed as previously described Chen et al. [64]. The yeast motif library was incubated overnight, after which an Ex-Yeast Transformation Kit was utilized to generate competent cells. Then, the pGADT7-AcDOF4 was transformed into competent yeast library cells. The yeast was transferred to plates of SD/-His/-Trp/-Leu media supplemented with 60 mM 3-AT and allowed to grow at 28 °C for five days. Monoclonal colonies were selected for sequencing, and random motif sequences between “GGG” and “CCC” (the SmaI site) were screened. The insertion sequences were analyzed using PlantCARE3 to identify whether they were known motifs.

## 5. Conclusions

Fruitlet abscission is a key limiting factor for the *A. catechu* industry. The *AcDOF* gene family was proven to be a critical regulator in the fruitlet abscission process. Six *AcDOF* genes with significant up-regulation expression levels in AZ during fruitlet abscission were identified based on the transcriptome data. The expression patterns of these *AcDOF* genes were positively correlated with the fruitlet abscission process. Through the screening in the whole *A. catechu* genome, a total of 36 *AcDOF* genes were identified and classified into nine subgroups, a total of nine types of plant hormone response *cis*-elements, and five types of abiotic stress related *cis*-elements were identified in the promoter regions of the *AcDOF* genes. Histochemical staining showed that the lignification degree of vascular bundles in AZ was much lower than that in pedicel and mesocarp, which might be a critical process for AZ formation. Our results suggested that the DOF transcription factor plays an important role in fruitlet abscission in *A. catechu*.

## Figures and Tables

**Figure 1 ijms-23-11768-f001:**
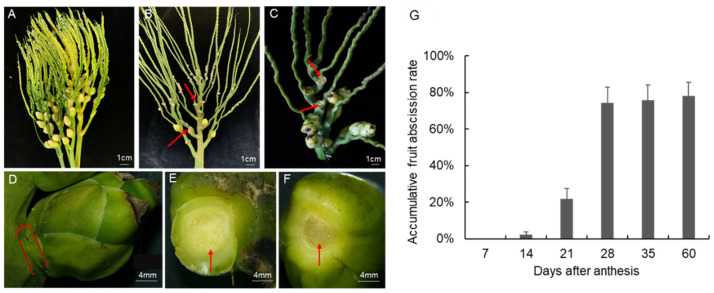
Morphological changes of *A. catechu* fruitlet during abscission. (**A**), Inflorescences with bract dehiscence; (**B**), Inflorescences with pistillate flowers abscised; (**C**), Inflorescences with fruitlet abscised; (**D**), non-abscised fruitlet; (**E**), pedicel part with fruitlet abscised; (**F**), fruit part of an abscised fruitlet; (**G**), Field data of accumulative fruitlet abscission rate in *A. catechu*. Red arrows represent the abscission zone.

**Figure 2 ijms-23-11768-f002:**
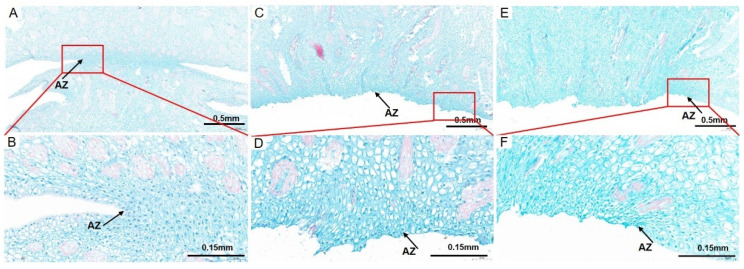
The cell morphological analysis of AZ and adjacent area in *A. catechu* fruitlet. Longitudinal sections of the *A. catechu* fruit bases were stained with Safranine O-Fast green FCF. (**A**), a partially abscised AZ and adjacent area; (**B**), the enlarged view of the red square area in (**A**); (**C**), the abscised AZ parts of the fruitlet; (**D**), the enlarged view of the red square area in (**C**); (**E**), the abscised AZ parts of the pedicel; (**F**), the enlarged view of the red square area in (**E**). The black arrow refers to the AZ cells.

**Figure 3 ijms-23-11768-f003:**
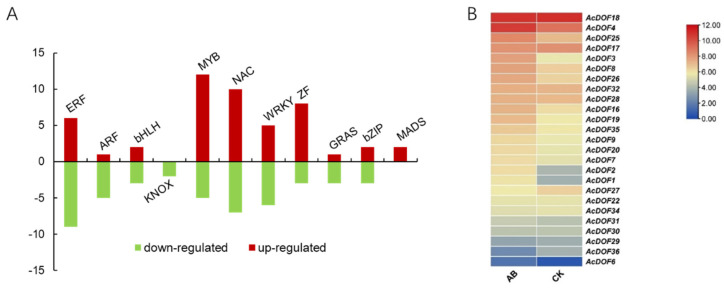
Differentially expressed genes encoding transcription factor during abscission. (**A**). Summary of the number of significant changes in transcription factors between the different families. (**B**). Expression profilings of the *AcDOF* genes in the fruitlet AZ. CK, the “non-abscised” part of the fruitlet AZ; AB, the “about-to-abscise” part of the fruitlet AZ.

**Figure 4 ijms-23-11768-f004:**
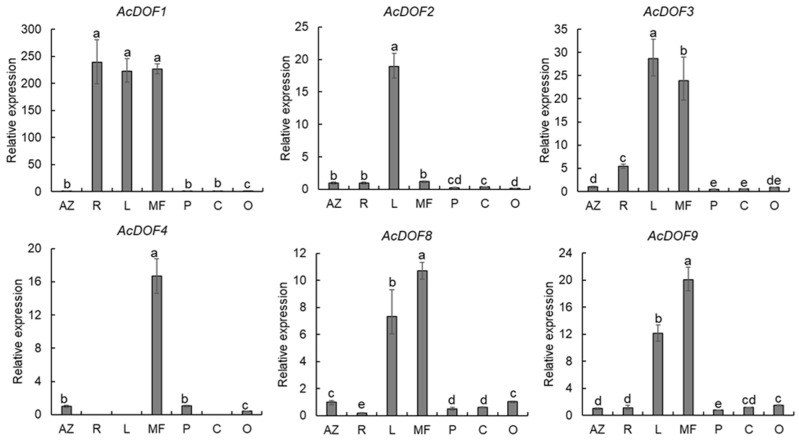
Expression profiles of different expressional *AcDOF* genes in various tissues determined by qRT-PCR. AZ, abscission zone; R, roots; L, leafs; MF, male flowers; P, Petals; C, calyxs; O, Ovary. Different letters above the bars stand for significant differences (Tukey’s multiple range tests, *p* < 0.05).

**Figure 5 ijms-23-11768-f005:**
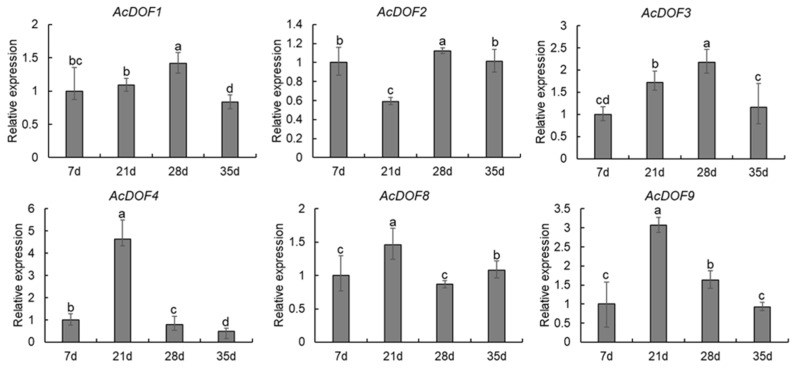
Expression profiles of differentially expressed *AcDOF* genes determined by qRT-PCR during fruitlet abscission. Different letters above the bars indicate significant differences (Tukey’s multiple range tests, *p* < 0.05) between different fruitlet abscission times.

**Figure 6 ijms-23-11768-f006:**
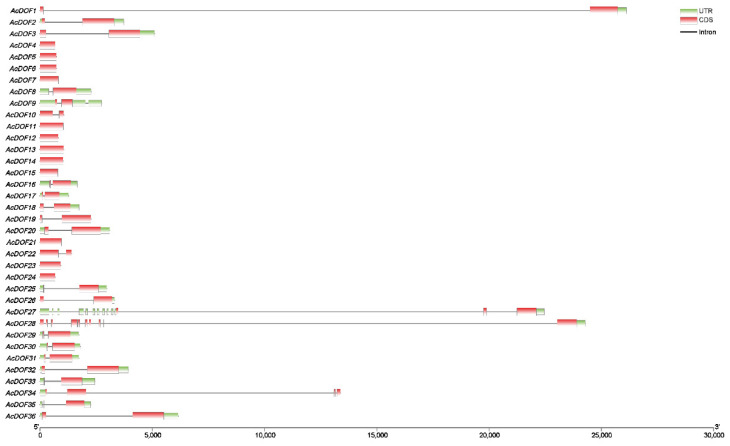
Intron-exon organization of 36 DOF transcription factors in the *A. catechu* genome. Coding sequences (CDS) are represented by red color blocks; 3′ & 5′ un-translated (UTR) regions are represented by green color blocks; intron regions are represented by grey color lines.

**Figure 7 ijms-23-11768-f007:**
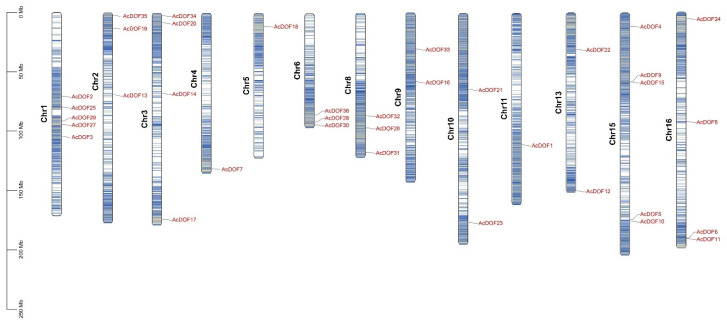
The *AcDOF* genes distribution across 13 chromosomes of *A. catechu* genome. Chromosome numbers are shown on the left side of each chromosome. The scale represents the length of *A. catechu* chromosomes.

**Figure 8 ijms-23-11768-f008:**
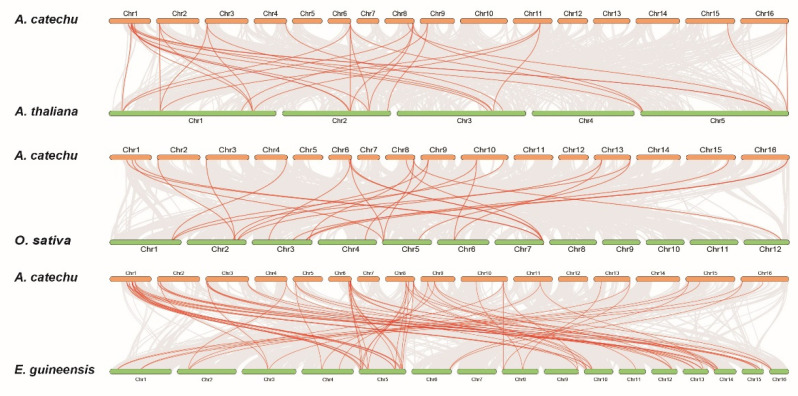
Collinearity analysis of the *DOF* gene family in monocot and dicot species.

**Figure 9 ijms-23-11768-f009:**
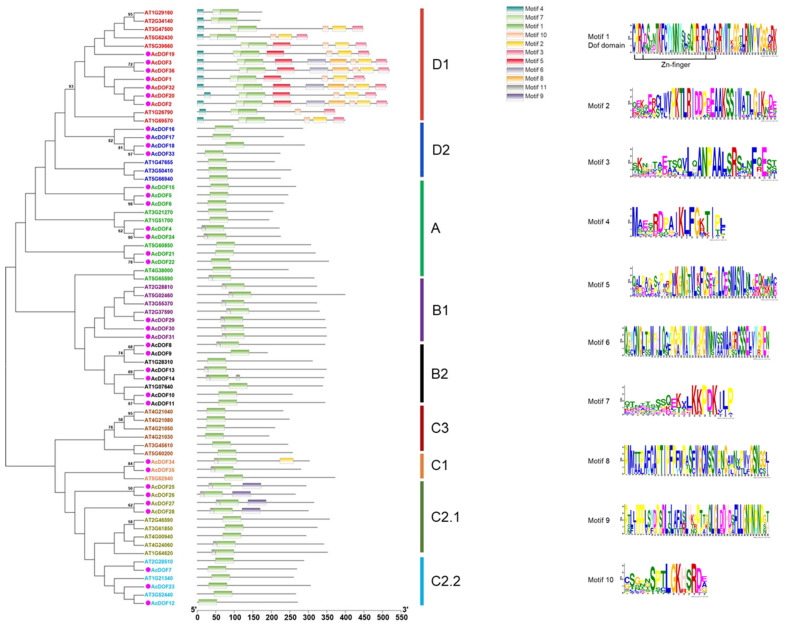
Phylogenetic tree and distribution of conserved motifs for the *Arabidopsis* and *A. catechu* DOF proteins. Ten motifs were identified in DOF protein using MEME tool. Each motif in DOF proteins is represented with different colors. The abundance of each amino acid in every motif of *A. catechu* DOF proteins is given in the sequence logo.

**Figure 10 ijms-23-11768-f010:**
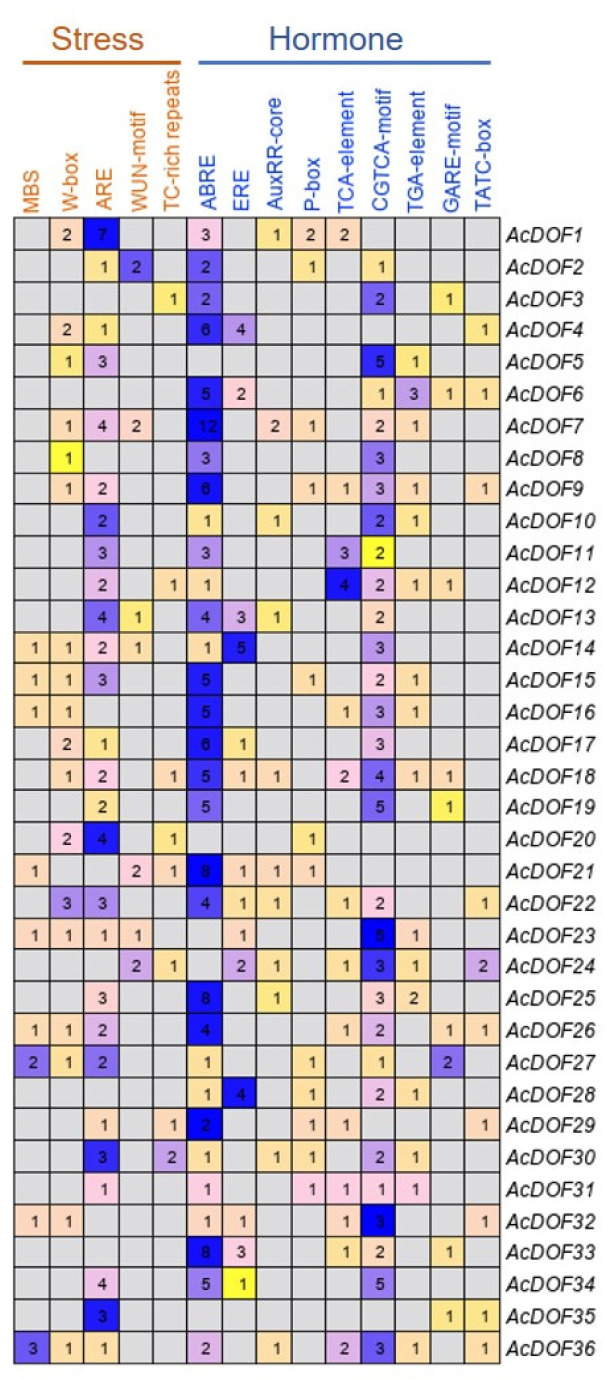
Distribution of stress- and hormone-related *cis*-elements in the promoter regions of the *AcDOF* genes. Each digit represents the number of a type of cis-element existing in the promoter region of the corresponding gene.

**Figure 11 ijms-23-11768-f011:**
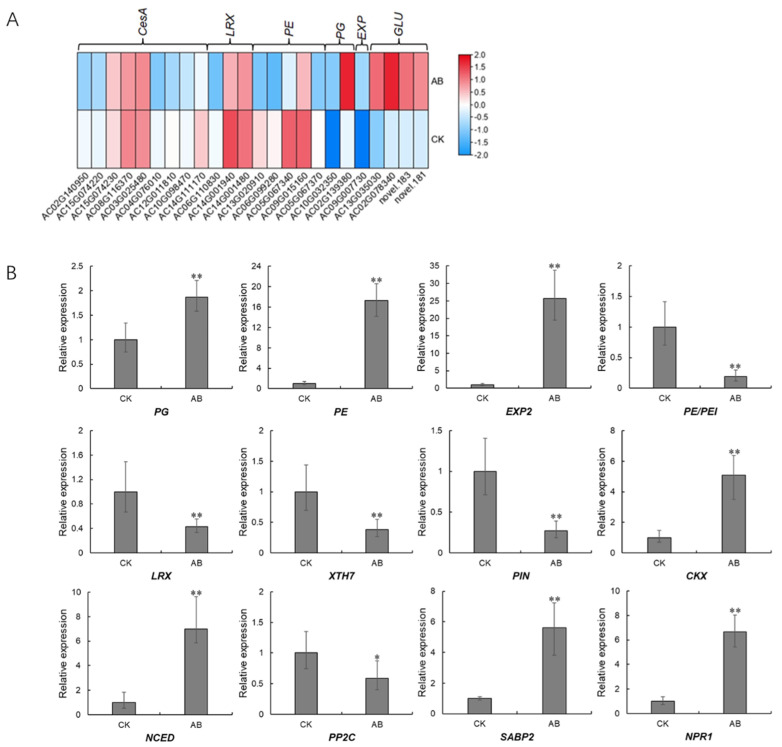
Expression patterns of downstream genes involved in abscission. (**A**), expression profiles derived from the transcriptome data; (**B**), expression profiles verified by qPCR. CK, the “non-abscised” parts of the fruitlet AZ. Asterisks indicate statistically significant differences compared with the CK (Student’s t-test: * *p* < 0.05; ** *p* < 0.01); AB, the “about-to-abscise” parts of the fruitlet AZ.

**Table 1 ijms-23-11768-t001:** Summary of *AcDOF* sequences and the identities of likely *A. thaliana* homologs.

Gene Name	Gene ID	Map Position (bp)	Amino Acids Length (aa)	AtDOF Homologs	Locus Name	pI	MW	Subcellular Localization
*AcDOF1*	Acat_11g009250	Chr11:110699065–110724793	452	*AtCDF2*	AT5G39660	6.65	49,350.83	N
*AcDOF2*	Acat_1g004830	Chr1:70534776–70537996	513	*AtCDF2*	AT5G39660	5.87	55,246.43	N
*AcDOF3*	Acat_1g014570	Chr1:104561992–104566442	512	*AtCDF2*	AT5G39660	5.17	55,134.50	N
*AcDOF4*	Acat_15g002830	Chr15:11111749–11112414	221	*AtDOF1*	AT1G51700	8.54	23,222.67	N
*AcDOF5*	Acat_15g024390	Chr15:173975247–173975981	244		AT5G66940	8.86	25,414.55	N
*AcDOF6*	Acat_16g018180	Chr16:190383661–190384383	233		AT5G66940	7.58	24,335.43	N
*AcDOF7*	Acat_4g014130	Chr4:130948723–130949541	268		AT3G52440	5.47	29,699.67	N
*AcDOF8*	Acat_16g012890	Chr16:92151393–92152427	344		AT1G28310	8.72	36,806.42	N
*AcDOF9*	Acat_15g016800	Chr15:58050539–58051318	189		AT1G28310	9.96	20,612.62	N
*AcDOF10*	Acat_15g024500	Chr15:175155782–175156840	256		AT5G65590	9.03	26,729.85	N
*AcDOF11*	Acat_16g018260	Chr16:190741627–190742661	344	*AtTMO6*	AT5G60200	8.71	36,713.22	N
*AcDOF12*	Acat_13g018160	Chr13:149838041–149838853	270		AT3G52440	7.08	29,677.29	M
*AcDOF13*	Acat_2g008700	Chr2:69123039–69124085	348		AT5G65590	8.70	37,279.17	N
*AcDOF14*	Acat_3g010220	Chr3:67580706–67581731	341		AT5G65590	8.84	36,788.77	N
*AcDOF15*	Acat_15g016820	Chr15:58158766–58159563	266	*AtTMO6*	AT5G60200	8.55	28,680.26	N
*AcDOF16*	Acat_9g016830	Chr9:58133888–58134838	284	*AtTMO6*	AT5G60200	8.71	31,008.63	N
*AcDOF17*	Acat_3g017820	Chr3:173350131–173350908	232	*AtDOF4.7*	AT4G38000	6.20	25,458.22	N
*AcDOF18*	Acat_5g003910	Chr5:10840469–10841812	289		AT1G28310	5.80	31,809.53	N
*AcDOF19*	Acat_2g004270	Chr2:12915981–12918246	464	*AtCDF2*	AT5G39660	8.90	50,944.69	N
*AcDOF20*	Acat_3g002830	Chr3:8090519–8093016	483	*AtCDF3*	AT3G47500	6.17	52,764.40	N
*AcDOF21*	Acat_10g017840	Chr10:64016604–64017560	318	*AtOBP4*	AT5G60850	9.06	32,774.20	N
*AcDOF22*	Acat_13g009530	Chr13:30724224–30725620	354	*AtOBP4*	AT5G60850	6.98	37,054.30	N
*AcDOF23*	Acat_10g024650	Chr10:176050006–176050923	305		AT3G52440	6.13	33,585.65	N
*AcDOF24*	Acat_16g001860	Chr16:5417374–5418048	224	*AtDOF1.7*	AT1G51700	7.59	23,389.84	N
*AcDOF25*	Acat_1g007750	Chr16:5417374–5418048	293		AT2G28510	8.06	31,484.15	N
*AcDOF26*	Acat_8g010760	Chr8:96494142–96497353	264	*AtDOF6*	AT3G45610	8.76	28,180.56	N
*AcDOF27*	Acat_1g011240	Chr1:94327689–94346398	314		AT2G28510	8.77	33,913.06	N
*AcDOF28*	Acat_6g009940	Chr6:91080639–91104544	299	*AtDOF6*	AT3G45610	7.61	32,493.28	M
*AcDOF29*	Acat_1g010000	Chr1:91336713–91337938	344	*AtOBP3*	AT3G55370	9.41	36,006.20	N
*AcDOF30*	Acat_6g010920	Chr6:93565735–93566971	347	*AtOBP3*	AT3G55370	9.35	35,942.32	N
*AcDOF31*	Acat_8g017000	Chr8:117065979–117067225	348	*AtDOF2.4*	AT2G37590	9.17	35,838.97	N
*AcDOF32*	Acat_8g007340	Chr8:86293841–86297264	510	*AtCDF3*	AT3G47500	5.79	55,454.90	N
*AcDOF33*	Acat_9g009220	Chr9:30488126–30489822	223	*AtDOF2.4*	AT2G37590	6.98	24,431.27	N
*AcDOF34*	Acat_3g000730	Chr3:2189612–2202743	302	*AtHCA2*	AT5G62940	6.85	32,815.97	N
*AcDOF35*	Acat_2g000910	Chr2:2239027–2240827	279	*AtHCA2*	AT5G62940	7.16	30,714.29	N
*AcDOF36*	Acat_6g007710	Chr6:85201346–85206767	518	*AtCDF3*	AT3G47500	5.16	56,239.77	N

PI, isoelectric point; MW, molecular weight; N, nucleus; M, mitochondrial.

**Table 2 ijms-23-11768-t002:** Distribution of motifs recognized by AcDOF4 in the promoter regions of genes involved in abscission.

Sequence	Motif	Genes													
		*AcCesA5*	*AcCesA5*	*AcCslD2*	*AcCesA7*	*AcCslD2*	*AcCesA2*	*AcLRX4*	*AcLRX4*	*AcLRX7*	*AcPE/PEI12*	*AcPE/PEI34*	*AcPG*	*AcEXP2*	*GLU*
+CTGGTCC −GACCAGG	Unnamed								1						
+CGGGGCC −GCCCCGG	Unnamed	1										1		1	
+GGGGGCC −CCCCCGG	Unnamed													1	
+GGAGGGC −CCTCCCG	Unnamed		1	2											
+CCGGGGC −GGCCCCG	Unnamed	2													
+GGGGCGG −CCCCGCC	Sp1		1					1							
+CGGCGGC −GCCGCCG	Unnamed	1						1				3			
+TAACGCC −ATTGCGG	Unnamed				1										
+CGTCCGC −GCAGGCG	Unnamed					1				1					
+TAGCTGC −ATCGACG	Unnamed			1											
+TGTCGGC −ACAGCCG	DRE core	1	1												
+CGCGTGG −GCGCACC	Unnamed										1				
+CGTCGGC −GCAGCCG	DRE core											1			
+GACCTGG −CTGGACC	Unnamed												1		
+TGTCTCT −ACAGAGA	Unnamed			1			1	1	1	1					
+CGGCTGC −GCCGACG	Unnamed														1
+CGCCGGG −GCGGCCC	Unnamed	1					1	1		1		2			1
+CGGCCCC −GCCGGGG	Unnamed									1					
+TGGCGGC −ACCGCCG	Unnamed	1						3	1						
Total	Unnamed	7	3	4	1	1	2	7	3	4	1	7	1	2	2

## Data Availability

Not applicable.

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
