# Peer review of "Genome-Wide Identification of the DOF Gene Family Involved in Fruitlet Abscission in Areca catechu L."

_ijms, 2022, doi:10.3390/ijms231911768_

Round 1

Reviewer 1 Report

Abscission, not only fruit, is a problem that cannot be ignored in commercial crops, so the research work in this article is definitely meaningful. The authors identified 36 DOF gene family members in the Areca catechu genome and analyzed the expression level of these members in fruitlet abscission. And provide some morphological and cytological observations of the abscission zone of fruitlet fruit. However, this article still needs some further revisions. Manuscripts can be published with major revisions.

1. line 21. Please change "plant hormone" to " Phytohormone".

2. line 21. "cis" should be italicized. Line 24-26. In terms of the content of the manuscript, the authors did not do an in-depth mechanistic analysis, just expression and cytological observations. The expression is also not RNA in situ hybridization, nor does it involve the regulation of downstream genes, etc. So, please note the wording.

3. Throughout the content arrangement of the author in the manuscript, the abstract should first introduce the cell structure, followed by the analysis of the gene family.

4. line 51. Please change “will shed at the fruitlet stage” to “will shed during the fruitlet development period”

5. line 50-59. This whole paragraph is too abrupt here. Since the authors mentioned DOF genes above, they should just continue to introduce them below. Please ask the authors to place this paragraph in the right place.

6. line 66. Please change “cys” to “Cys”, Please check the full text.

7. line 68-70. Pay attention to punctuation. Please describe them in one sentence.

8. Zinc finger proteins have been studied more in plants, and DNA binding ability is an important feature of transcription factors. The authors can try to predict the DNA binding capacity of the DOF family, but it is not necessary. However, in the future, the authors can consider these works when conducting in-depth research.

9. line 85-87. Write these two sentences into one sentence.

10. line 154. Note the format of log2Fold.

11. The author selected 6 genes for qRT-PCR. If based on RNA-seq, some genes such as DOF16/19/25 are differential, why only these 6 genes were selected. In addition, DOF18 is highly expressed in both samples, how do the authors explain this phenomenon?

12. In Table 1, the PI should be changed to pI.

13. In the caption to Figure 6, the authors should explain what the different colors in the chromosomes represent.

14. It is recommended to combine Figures 7 and 8 into a single figure.

15. In the phylogenetic tree in Figure 8, why are some locations missing bootstrap values?

16. The authors are missing an analysis of the intron-exon structure in the manuscript. This part of the work is relatively well done, and the authors are requested to add the analysis of the gene structure in the manuscript. Because through the gene structure, the conservation of this gene family can be understood. In addition, the number of introns may be related to gene expression, and the conservation of introns may also be related to the evolution of gene families.

17. The authors analyzed the cis-acting element of the upstream sequence of DOF. The authors were asked to combine the results of this analysis with qRT-PCR to analyze the gene expression of this gene family under different hormone or abiotic stress treatments.

18. Don't the authors feel they have done too little research? This clearly does not meet the requirements for publication. The authors did only a few gene expression and paraffin sections, and the experimental data is too little. It is recommended that the authors add an analysis of gene family collinearity and duplication events in the Gene Family Analysis section. Importantly, authors should flesh out their manuscripts with additional gene expression data. Alternatively, through genetic transformation, provide genetic evidence that these genes are associated with abscising.

19. line 313-319. This paragraph is a repetition of the results.

20. line 400. Please provide more details.

Author Response

Point to point response

  1. line 21. Please change "plant hormone" to " Phytohormone".

Thanks. The phrase "plant hormone" has been replaced by "phytohormone" in all texts of this manuscript.

  1. line 21. "cis" should be italicized. Line 24-26. In terms of the content of the manuscript, the authors did not do an in-depth mechanistic analysis, just expression and cytological observations. The expression is also not RNA in situ hybridization, nor does it involve the regulation of downstream genes, etc. So, please note the wording.

Thanks for pointing out this error. The word "cis" has been replaced by "cis" in all texts of this manuscript.

The sentence in line 24-26 has been rewritten.

  1. Throughout the content arrangement of the author in the manuscript, the abstract should first introduce the cell structure, followed by the analysis of the gene family.

Thanks. The content with respect to the cell structure has been added into the abstract.

  1. line 51. Please change “will shed at the fruitlet stage” to “will shed during the fruitlet development period”

Thanks. The sentence has been revised according to the comment.

  1. line 50-59. This whole paragraph is too abrupt here. Since the authors mentioned DOF genes above, they should just continue to introduce them below. Please ask the authors to place this paragraph in the right place.

Thanks. This content has been placed at the beginning of the introduction to make it more logical.

  1. line 66. Please change “cys” to “Cys”, Please check the full text.

Thanks. The word “cys” has been replaced by “Cys” in the full text.

  1. line 68-70. Pay attention to punctuation. Please describe them in one sentence.

Thanks. These sentences have been rewritten.

  1. Zinc finger proteins have been studied more in plants, and DNA binding ability is an important feature of transcription factors. The authors can try to predict the DNA binding capacity of the DOF family, but it is not necessary. However, in the future, the authors can consider these works when conducting in-depth research.

Thanks for this suggestion. Actually, we performed the prediction through a TF (transcription factor)-centered Yeast One Hybrid technique to identify cis-elements that can be recognized and bound by AcDOF4. Totally 33 cis-element motifs were identified and some of them indicated the regulatory relationship between DOF and downstream genes involved in abscission, such as genes encoding cellulose synthase A catalytic subunit 5, pectinesterase/pectinesterase inhibitor 41, expansin 2, leucine-rich repeat extensin-like protein 7 and cellulose synthase-like protein. We have added this content into the results.

  1. line 85-87. Write these two sentences into one sentence.

Thanks. These sentences has been rewritten.

  1. line 154. Note the format of.

Thanks. The format “log2Fold” has been revised to “log2Fold”.

  1. The author selected 6 genes for qRT-PCR. If based on RNA-seq, some genes such as DOF16/19/25 are differential, why only these 6 genes were selected. In addition, DOF18 is highly expressed in both samples, how do the authors explain this phenomenon?

Totally 25 AcDOF genes were specifically expressed in AZ, and 6 of them showed significantly different expression (False Discovery Rate (FDR) < 0.05 and | log2 Fold Change| ≥ 1) between “about-to-abscise” and “non-abscised” parts of AZ, including AcDOF1, AcDOF2, AcDOF3, AcDOF4, AcDOF8 and AcDOF9. Therefore, these 6 genes were selected for qPCR analysis. The DOF16/19/25 genes were also identified as DEGs, however, the FDR values of them were higher than 0.05, which means that the difference of them is not statistically significant. Therefore, these genes were not selected.

For the high expression of the DOF18 gene in both samples, we speculated that this gene is necessary for fruit development but not specifically involved in abscission. The different expression patterns among the DOF genes indicates fine division in this family.

  1. In Table 1, the PI should be changed to pI.

Thanks for pointing out this error. The “PI” has been replaced by “pI”.

  1. In the caption to Figure 6, the authors should explain what the different colors in the chromosomes represent.

The different color represents the density of gene arrangement in the chromosome. This information has been added into the figure caption of Figure 6.

  1. It is recommended to combine Figures 7 and 8 into a single figure.

Thanks. We have combined Figures 7 and 8 into a new figure (Figure 9 in the revision version).

  1. In the phylogenetic tree in Figure 8, why are some locations missing bootstrap values?

We set the parameter of bootstrap value to be visible when it was more than 50. Therefore, some locations will not show bootstrap values if the value is less than 50 when the phylogenetic tree was generated.

  1. The authors are missing an analysis of the intron-exon structure in the manuscript. This part of the work is relatively well done, and the authors are requested to add the analysis of the gene structure in the manuscript. Because through the gene structure, the conservation of this gene family can be understood. In addition, the number of introns may be related to gene expression, and the conservation of introns may also be related to the evolution of gene families.

We have added graphics of the gene structure including intron-exon analysis in the result section, and we also added corresponding content in the discussion section.

  1. The authors analyzed the cis-acting element of the upstream sequence of DOF. The authors were asked to combine the results of this analysis with qRT-PCR to analyze the gene expression of this gene family under different hormone or abiotic stress treatments.

Thanks. The relationship between cis-elements and gene expression patterns has been further discussed.

  1. Don't the authors feel they have done too little research? This clearly does not meet the requirements for publication. The authors did only a few gene expression and paraffin sections, and the experimental data is too little. It is recommended that the authors add an analysis of gene family collinearity and duplication events in the Gene Family Analysis section. Importantly, authors should flesh out their manuscripts with additional gene expression data. Alternatively, through genetic transformation, provide genetic evidence that these genes are associated with abscising.

Thanks for this suggestion. We have added the collinearity analysis of the DOF gene family from different species into the results.

  1. line 313-319. This paragraph is a repetition of the results.

Thanks. The repetition has been deleted.

  1. line 400. Please provide more details.

Thanks. Detailed information has been added.

Reviewer 2 Report

The authors of the current paper perform a genome-wide analysis of Dof TFs approach and explain their potential role in fruit abscission. To back up this theory, the authors conducted morphological analysis and correlated their expression pattern with fruit abscission. The manuscript appears to be lacked enough data and it requires significant revision before it can be accepted. Here are some of my concerns/suggestions that should be addressed.

·        The authors stated that they performed RNA-seq data, but I was unable to find any data relating to this section. I recommend that RNA-seq data be included in the manuscript to improve its quality.

·        Could you explain the difference that this abscission data is correct and not induced by external stimuli in the Material and Method section, author mentioned that they perform this assay in 2020-2021, as there is no repetition?

·        In the phylogenetic analysis, the authors use ML rather than the NJ method; is there a specific reason or reference for this methodology choice?

·        The authors discover gene expression after 7, 21, 28, and 35 days; how about 14 days of data?

·        The legends for the figures are too short and require more explanation.

·        Instead of single digits, the Cis-elements should be shown in technical hormone terminology such as GA, MeJA, ABA, Aux, SA.

·        In the manuscript, Cis-element should be changed into Cis-element.

·        I suggest to keep the single name of plant species, either botanical or common name

·        The full abbreviations for Aux/IAA, ERF, and MADS-box are required.

The manuscript contains a single gene family analysis, which they did not analyse thoroughly, which is the manuscript's weakest point. In its current state, the manuscript is unacceptable.

Author Response

  1. The authors stated that they performed RNA-seq data, but I was unable to find any data relating to this section. I recommend that RNA-seq data be included in the manuscript to improve its quality.

Thanks. The transcriptome data has been deposited into China National Center for Bioinformation with the code CRA007290. Related information is present in another our manuscript which is under submission. To support our finding that the AcDOFs are closely related to fruitlet abscission, partial transcriptome data with respect to the differentially expressed genes encoding transcription factors has been added into the results.

  1. Could you explain the difference that this abscission data is correct and not induced by external stimuli in the Material and Method section, author mentioned that they perform this assay in 2020-2021, as there is no repetition?

Thanks. Totally 30 individuals grown in the planting area were randomly selected to perform the statistics of fruitlet abscission rate, and this assay was repeatedly performed during 2020-2021 to obtain the means of abscission rate. External stimuli might induce abscission, while the number of fruits we collected during sampling was far less than the number of fruits yielded on a tree. Therefore sampling will hardly affect the statistics of abscission rate.

  1. In the phylogenetic analysis, the authors use ML rather than the NJ method; is there a specific reason or reference for this methodology choice?

Actually, the ML and NJ method are both available for phylogenetic analysis. However, we found that long-branch attraction (LBA) often occurs when the NJ method is adopted, thus interfere the generation of phylogenetic trees. Therefore, we chose the ML method.

  1. The authors discover gene expression after 7, 21, 28, and 35 days; how about 14 days of data?

Thanks. During observation, we noticed that the fruitlet abscission was quite inconsistent among repetitions from 7 to 14 days, which might because that the fruitlet initiation is under precise control of physiological status, environmental cues and genetic factors. While after 21 days, the abscission rate among repetitions were much closer to each other.

  1. The legends for the figures are too short and require more explanation.

Thanks. The legends of all figures have been thoroughly revised to be more informative.

  1. Instead of single digits, the Cis-elements should be shown in technical hormone terminology such as GA, MeJA, ABA, Aux, SA.

Thanks. We have revised related information.

  1. In the manuscript, Cis-element should be changed into Cis-element.

Thanks for pointing out this error. The word "cis" has been replaced by "cis" in all texts of this manuscript.

  1. I suggest to keep the single name of plant species, either botanical or common name

Thanks for this suggestion. We carefully checked and revised the manuscript. We only use both botanical and common name of a species when it first appears in the text.

  1. The full abbreviations for Aux/IAA, ERF, and MADS-box are required.

The full names of Aux/IAA (Auxin/Indole-3-Acetic Acid), ERF (ethylene- responsive factor) and MADS-box (Mcml, Agamous, Deficiens and SRF 4) have been added into the text.

  1. The manuscript contains a single gene family analysis, which they did not analyse thoroughly, which is the manuscript's weakest point. In its current state, the manuscript is unacceptable.

Thanks for this suggestion. To flesh out our results, we added transcriptome data of transcription factors, intron-exon analysis, collinearity analysis and downstream gene prediction. We hope these revisions could improve the manuscript to be more convincing.

Round 2

Reviewer 1 Report

acceptable